# Development and Initial Validation of the PILCAST Questionnaire: Understanding Parents’ Intentions to Let Their Child Cycle or Walk to School

**DOI:** 10.3390/ijerph182111651

**Published:** 2021-11-06

**Authors:** Hanna Forsberg, Anna-Karin Lindqvist, Sonja Forward, Lars Nyberg, Stina Rutberg

**Affiliations:** 1Department of Health, Education and Technology, Luleå University of Technology, 971 87 Luleå, Sweden; anna-karin.lindqvist@ltu.se (A.-K.L.); lars.nyberg@ltu.se (L.N.); stina.rutberg@ltu.se (S.R.); 2Swedish Road and Transport Research Institute, VTI, 581 95 Linköping, Sweden; sonja.forward@vti.se

**Keywords:** active school transportation, active commuting, children, parents, theory of planned behavior, intentions, school setting

## Abstract

Children generally do not meet the recommendation of 60 min of daily physical activity (PA); therefore, active school transportation (AST) is an opportunity to increase PA. To promote AST, the involvement of parents seems essential. Using the theory of planned behavior (TPB), the aim was to develop and validate the PILCAST questionnaire to understand parents’ intentions to let their child cycle or walk to school. Cross-sectional sampling was performed, where 1024 responses were collected from parents. Confirmatory factor analysis indicated acceptable fit indices for the factorial structure according to the TPB, comprising 32 items grouped in 11 latent constructs. All constructs showed satisfying reliability. The regression analysis showed that the TPB explained 55.3% of parents’ intentions to let the child cycle to school and 20.6% regarding walking, increasing by a further 18.3% and 16.6%, respectively, when past behavior was added. The most influential factors regarding cycling were facilitating perceived behavioral control, positive attitudes, subjective and descriptive norms, and for walking, subjective and descriptive norms. The PILCAST questionnaire contributes to a better understanding of the psychological antecedents involving parents’ decisions to let their child cycle or walk to school, and may therefore provide guidance when designing, implementing and evaluating interventions aiming to promote AST.

## 1. Introduction

Children’s declining levels of physical activity (PA) are becoming a major threat to their health worldwide, and interventions are needed to increase the possibilities to meet the WHO recommendations of 60 min daily PA [1,2,3]. Active school transportation (AST), also known as cycling and walking to school, is an opportunity to increase PA at the population level [4,5]. Unfortunately, only about 54–59% of children and the youth in Sweden use active transport modes [6]. This is low compared to neighboring countries such as Finland and Denmark, where 74–79% of children and the youth use active transport. PA behavior, including active transport, tends to form in younger years and develop into adulthood [7,8]. Thus, efforts should be directed in childhood when aiming to enhance long-term health improvements. To promote AST, reviews have highlighted the need to involve parents, because they are the primary decision makers of children’s AST [9].

Previous studies have shown that parents’ perceptions, attitude, social support and perceived barriers towards AST influence their children’s transport mode to school [9,10,11,12,13]. The barriers most frequently reported by parents are traffic safety, the built environment and distance to school [9]. In addition, research has shown that there are more parental concerns regarding cycling compared to walking, in terms of traffic safety [10]. In many developed countries such as the United Kingdom [14] and Spain [15], walking to school is a more common form of AST; meanwhile, cycling is more common in countries such as Denmark, The Netherlands and Germany [16,17]. These differences have been suggested to be related to different walking and cycling cultures, social norms and infrastructure [18]. Reviews also stress the need for more studies considering these behaviors (i.e., cycling and walking to school) separately, and more knowledge is especially needed around cycling to school [19,20]. Substantial efforts have been dedicated to the understanding of parental barriers towards AST, providing valuable knowledge [9,19]. Nevertheless, there is also a need to focus on factors that parents perceive as facilitating to promote behavior changes regarding AST [19]. To do this, researchers are urged to ground their investigations in theoretical foundations [19].

### Theoretical Framework

The theory of planned behavior (TPB) has successfully been used in previous efforts to understand AST behavior from a parental perspective [21,22,23]. Briefly, the TPB suggests that personal decisions (intentions) are based on a combination of attitudes toward the behavior, subjective norms, and perceived behavioral control [24]. These constructs are also described as direct measures that are determined by three belief-based (indirect) measures; behavior beliefs, normative beliefs and control beliefs. The difference between direct measures and indirect measures is that direct measures focus directly on the concept in question, whereas indirect measures focus on the presumed determinants from which the concept can be inferred [25]. Belief-based measures are of great interest in research because they provide a deeper understanding of what motivates a person’s decision [26]. Demographic factors such as age and gender are, according to the theory, more distal predictors of the behavior influencing the individuals’ beliefs indirectly [24].

However, subjective norms have been argued to be too narrow a concept of norms, and that additional norms should therefore be included such as descriptive norms [27]. Descriptive norms are different from subjective norms because they refers to what is done, rather than something that should be done. Previous meta-analyses have confirmed descriptive norms to be successful variables in efforts to understand behavior by increasing the variance, with approximately 5% above the other constructs in the TPB [27].

Behaviors are carried out with little effort when attitudes become more established [28]. The behavior will then be persistent until something challenges the motive of a person’s decision. At this stage, the behavior could be considered as more or less habituated. Habit is not included in the TPB, but has previously been added to understand different behaviors [29], including children’s AST [30]. The measure of habit has varied, and is sometimes treated similar to past behavior, because things we do often have a tendency to become habituated [28]. Children’s school travel is a routine behavior performed repeatedly during each school day, and parents’ intentions (decisions) can therefore not be precluded from being under the influence of habituation [30,31]. Finally, to the best of our knowledge, very few studies have used the TPB to understand parents’ intentions regarding AST [21,22,23]. None of the previous studies have included indirect (belief-based) measures, and have only covered parents’ intentions regarding their child walking to school [21,22,23]. Therefore, to address the gap of knowledge and to increase the potential of understanding parents’ intentions to let their child cycle or walk to school, the aim of this study was to develop and validate a questionnaire using the TPB as a framework, called PILCAST (parents’ intentions to let their child use AST).

## 2. Methods

### 2.1. Procedure and Measures

The questionnaire was based on an extended version of the TPB [27,32] and developed in *four* phases. In the *first* phase of the development, parents’ salient behavioral beliefs were obtained with the aid of a qualitative study using individual semi-structured interviews [33].

In the *second* phase (a), in line with the guidelines of Ajzen [32], the most common and frequent assertions from the qualitative study [33] were formed into statements by the research team. The research team consisted of experts with extensive knowledge of children’s school travel and parents’ attitudes towards AST (authors 1, 2 and 5), in addition to one researcher with extensive knowledge of developing questionnaires based on the TPB (author 3). To ensure that all parents responded to the defined behavior according to TACT (target, action, context and time), a scenario was added to the questionnaire [26,32] that asked the respondents to imagine themselves in the following depicted situation: *“Imagine, your child cycling or walking to school this time a year when the weather is nice and clear. Please, try to consider the following statements even if your*
*child does not cycle or walk to school”*. Based on the assumption that cycling and walking to school are different behaviors [19,20], they were treated separately in the questionnaire.

In the *second* phase (b), the formed statements measured the theoretical constructs of the TPB, which covered intention, attitudes, subjective norm, descriptive norm and perceived behavioral control [32]. Positive and negative attitudes were assessed using a combination of behavioral beliefs (BBs) that refers to the consequences of the behavior and outcome evaluations (OEs), referring to evaluations of those consequences. Each BB question started with “*If you would let your child cycle/ walk to school, how much would you agree to the following statements?*” The answer options ranged from 1 = Strongly disagree, to 7 = Strongly agree. OE consisted of the same disposal of beliefs, but was rephrased on how important they were (1 = Not very important, to 7 = Very important).

Two types of social norms were included: subjective norms (SNs) and descriptive norms (DNs). Each question regarding SNs started with “*What do you think others in your immediate vicinity would think about you, letting your child cycle/ walk to school?*” (1 = Completely unacceptable, to 7 = Completely acceptable, and “I do not know/Does not apply to me”). Each question for DNs started with *“If you consider those in your immediate vicinity who have children under the age of twelve, do they let their children cycle/ walk to school?*” (1 = Strongly disagree, to 7 = Strongly agree and “I do not know/Does not apply to me”).

Impeding and facilitating perceived behavioral control were assessed with a combination of control belief strength (CBS), which is the likelihood of the factor being present, and control belief power (CBP), which is the perceived power of these factors. Each question for CBS started with “*How much would the following impede/facilitate you to let your child cycle/ walk to school*” (1 = Very little, to 7 = Very much). CBP questions consisted of the same disposal of beliefs, but were rephrased based on how this applied to their situation (Strongly disagree = 1, to Strongly agree = 7). Two questions assessed intention, and each question started with *“In the next three upcoming weeks I intend to”* and *“In the next three upcoming weeks, I plan to”* (1 = Strongly disagree, to 7 = Strongly agree).

In total, the questionnaire comprised 46 TPB belief-based measures grouped in 11 latent constructs, as well as 4 questions about the frequency of past behavior (cycling, walking, bus and car), for which the timeframe was set at the three previous weeks. Each question started with *“How many times during the past three weeks has your child “cycled”, “walked”, “used the bus” or been “driven by car” to school”*, with answer options ranging from 1 = approximately one day per week, to 6 = approximately five days a week. The last part of the questionnaire included socio-economic and demographic background questions. These questions included parents’ gender, age, ethnicity and education, and the child’s gender, academic school year and distance to school.

In the *third* phase, a web questionnaire was constructed and pilot-tested in a convenience sample [34] consisting of eleven parents (seven mothers and four fathers) living in both rural and suburban areas who had children in elementary school academic years 1–6. The process involved using the idea of the method “thinking aloud” [35], where the convenience sample was asked to fill out the questionnaire with the first author present. Throughout these sessions, participants were instructed to focus on clarity, readability, wording, formatting and missing answer options. At the end of each session, the first author summarized all of the participant views and checked that all of them were understood correctly. The interviews resulted in the removal of three items, changes to the wording, and adding and clarifying some questions and answer options.

In the *fourth* phase, a cross-sectional sampling design was applied, and the online questionnaire was administrated to parents through a school application by one communicator working in child and education administration. The web-based application is used by schools in the present municipality to enable information sharing to parents about school activities. The application could only be accessed using an electronic identification system [36], ensuring that only parents responded to the questionnaire. The questionnaire was published on the application in mid-September, four weeks after the school semester started. It was distributed to parents in all 30 municipal elementary school, in academic school year 1–6. If the parent had more than one child in elementary school, they were instructed to submit a unique questionnaire for each child. A reminder to respond to the web questionnaire was sent out after 14 days. Parents could respond to the web questionnaire for a total of 25 days. We aimed to have at least a 10:1 ratio for the 46 items, estimating a minimum of 460 responses. To enhance the willingness to respond, cinema tickets to the school classes with highest response rate of the web questionnaire were offered.

This study was conducted in line with ethical principles according to Swedish law for research and the World Medical Association’s Declaration of Helsinki [37]. The regional ethical review board in Umeå approved the study (Dr 2018–10-31 M). Parents were provided information about the purpose of the study, their voluntary participation and reassurance of confidentiality. Parents agreed to participate in the study by submitting the questionnaire.

### 2.2. Context

The studied municipality is located in the northern part of Sweden. Approximately 80,000 inhabitants live in the municipality, and over 19,000 people reside in the suburban and rural areas. The suburban and rural areas consist of smaller sparsely populated communities, and the city districts are more densely populated. The 30 schools included in the study are spread out in all of these mentioned areas. The climate within this area is characterized by long, cold, snowy winters and short springs, summers, and autumns [38]. The onset of autumn is normally in September and ends in November. During this time period, the ground is bare. There has been an ongoing intervention for the last few years in this region [39] to promote AST among school children.

### 2.3. Data Analysis

Construct validity was assessed with confirmatory factor analysis (CFA) using maximum likelihood estimation [29,40]. We followed Hairs et al.’s guidelines regarding goodness of fit (GOF) measures [41]. They suggest that more complex models with larger samples should not be held to the same strict GOF as those with smaller samples (large sample sizes > 1000) and fewer variables and less model complexity (>30 variables). The following fit indices were therefore used: the $x^{2}$ value and the associated *df*, one absolute fit index/badness of fit (GFI, RMSEA or SRMR); one incremental fit index (CFI or TLI); one goodness of fit (GFI, CFI or TLI); and one badness of fit (RMSEA or SRMR). Chi-squared ($x^{2}$), significant *p*-values expected, CFI or TLI > 0.92, RMSEA < 0.70 with CFI > 0.92, SRMR < 0.8 with CFI > 0.92. Significant standardized factor loadings of >0.4 were considered as acceptable [34]. As recommended when two models are compared, we reported the AIC value with lower values indicating a better model [42]. Factor loadings and modification indices were used to identify sources of misfit in the model [40]. Convergent validity was tested using the composite reliability coefficient of McDonald’s Omega with a value of >0.7 considered as acceptable, and average variance extracted (AVE) exceeding a value of >0.5 [41,43]. Discriminant validity was assessed using the criterion of Fornell and Larcker, where the squared root of each construct’s AVE was higher than the constructs correlation with another construct [44]. Descriptive statistics were used to display the socio-economic and demographic background characteristics of the participants. When construct validity had been assessed and prior to further analysis, in accordance with the guidelines of Ajzen indexes for all of the TPB constructs were summarized and combination scales were multiplied, forming composites [24]. Descriptive statistics and Pearson correlations were then calculated, and hierarchical multiple linear regression analysis was used to estimate parents’ intentions to let the child cycle or walk to school, with intention set as the dependent variable. The significance level was set at *p* < 0.05. Completion of the questionnaire was required for submission, and therefore there were no missing data. Answer options such as *“I do not know/Does not apply to me”* were treated as missing. Parents’ gender and children’s gender set as *“other”* were very few and were therefore treated as missing as well. Prior to the final analysis, two analyses were compared. In case one, missing values were replaced by variable means, but in the other, no imputation was made. The imputation of missing values by means did not change the overall interpretation of the results; thus, no imputation was made in the final analysis. All analyses were conducted using SPSS Statistics for Windows, Version 28.0. Armonk, NY, USA: IBM Corp and SPSS AMOS for Windows, 28.0 Chicago, IL, USA: IBM Corp. 

## 3. Results

### 3.1. Participants

Table 1 presents the participants’ socio-economic and demographic background characteristics. A sample of 1024 self-reported responses was collected from parents who had children in elementary school, academic school years 1–6. The sample comprised mostly higher educated (71.1%) women (79%) aged between 40 and 49 (51.9%). The children who formed the focus of the questionnaire were relatively evenly spread out in the academic school years (1–6). There was also an even split in terms of girls (46.9%) and boys (52.5%), except children who were reported as other (0.6%). A large proportion of the respondents (44.7%) in the sample lived close to school (<1.0 km) and originated from Sweden or the Nordic countries (95%).

**Table 1 ijerph-18-11651-t001:** Socio-economic and demographic background characteristics of the participants (*n* = 1024).

	%
**Gender of parent**	
Women	79.0
Men	20.9
Other	0.1
**Age of parent**	
18–29	2.2
30–39	37.9
40–49	51.9
>50	8.0
**Ethnicity of parent**	
Sweden and the Nordic countries	95.0
Non-Nordic countries	5.0
**Education of parent**	
Lower (elementary, secondary school or other)	28.2
Higher (higher education institution)	71.8
**Gender of child**	
Girl	46.9
Boy	52.5
Other	0.6
**Academic school year of child**	
Year 1	21.1
Year 2	16.6
Year 3	16.8
Year 4	16.1
Year 5	15.4
Year 6	14.0
**Distance to school (km)**	
0.0–1.0	44.7
1.1–2.0	28.2
2.1–3.0	13.1
3.1–4.0	4.4
4.1–5.0	2.1
5.1–10	4.1
>10	3.3

### 3.2. Construct Validity and Reliability

Table 2 provides an overview of the final values for the included TPB items displaying factor loadings, McDonald’s Omega and AVE values for each TPB construct. For the sake of clarity, the 11-factor solution divides behavior belief and outcome evaluation in positive (PBB and POE) and negative (NBB and NOE) factors. Likewise, control belief strength and control belief power were divided into facilitating factors (FCBS and FCBP) and impeding factors (ICBS and ICBP).

**Table 2 ijerph-18-11651-t002:** Overview of items, CFA factor loadings, McDonald’s Omega and AVE for the TPB cycling and walking constructs.

Items	Answer Options	Scale	TPB Cycling			TPB Walking		
			Factor Loading	ώ	AVE	Factor Loading	ώ	AVE
I intend to let my child I plan to let my child	1 = Strongly disagree 7 = Strongly agree	INT	0.991	0.977	0.955	0.973	0.968	0.938
			0.963			0.964		
Increased independency Improved concentration in school Improved health	1 = Strongly disagree 7 = Strongly agree	PBB	0.810	0.870	0.691	0.817	0.852	0.658
			0.812			0.801		
			0.870			0.816		
	1 = Not very important 7 = Very important	POE *	0.697	0.779	0.542	0.704	0.780	0.542
			0.809			0.808		
			0.696			0.692		
Too cumbersome preparations Trip takes too long	1 = Strongly disagree 7 = Strongly agree	NBB	0.733	0.812	0.686	0.696	0.802	0.674
			0.914			0.929		
	1 = Not very important 7 = Very important	NOE *	0.875	0.907	0.829	0.879	0.916	0.845
			0.945			0.958		
Friends Parents Coworker/fellow student	1 = Completely unacceptable 7 = Completely acceptable	SN	0.953	0.975	0.928	0.963	0.979	0.938
			0.954			0.963		
			0.982			0.980		
Friends Parents Coworker/fellow student	1 = Strongly disagree 7 = Strongly agree	DN	0.926	0.921	0.795	0.921	0.922	0.797
			0.874			0.887		
			0.874			0.870		
Crossing an unattended pedestrian crossing Crossing a major road Travel along roads with higher speeds than 40 km/h	1 = Very little 7 = Very much	ICBS	0.866	0.919	0.791	0.858	0.916	0.785
			0.966			0.965		
			0.831			0.829		
	1 = Strongly disagree 7 = Strongly agree	ICBP *	0.723	0.820	0.604	0.732	0.824	0.610
			0.818			0.807		
			0.788			0.802		
Trusting the child Child being able to navigate Safe environment Separate walking/cycling lanes	1 = Very little 7 = Very much	FCBS	0.867	0.858	0.606	0.905	0.870	0.632
			0.772			0.836		
			0.838			0.825		
			0.610			0.573		
	1 = Strongly disagree 7 = Strongly agree	FCBP *	0.903	0.817	0.546	0.887	0.796	0.514
			0.847			0.869		
			0.707			0.597		
			0.390			0.397		

Note: INT, intention; PBB, positive behavioral belief; POE, positive outcome evaluation; NBB, negative behavioral belief; NOE, negative outcome evaluation; SN, subjective norm; DN, descriptive norm; ICBS, impeding control belief strength; ICBP, impeding control belief power; FCBS, facilitating control belief strength; FCBP, facilitating control belief power; * subscale used in both models; ώ, McDonald’s Omega; AVE, average variance extracted.

The initial TPB cycling model comprising 46 items, grouped in 11 latent constructs, yielded the following results in the CFA: $x^{2}$ = 6568 df = 937 *p* < 0.05, CFI = 0.854, TLI = 0.838, SRMR = 0.061, RMSEA = 0.077, AIC = 6948. Factor loadings ranged from 0.239 to 0.991, showing that the model fit was inadequate. By following the standardized factor loadings and modification indices to improve fit, items were removed in a stepwise procedure [40]. In total, 14 items were removed, 12 of these were multi-composite items (3 items from the positive behavioral belief and positive outcome evaluation components, and 3 items from the impeding control belief strength component; correspondingly, 3 from the impeding control belief power component, 1 item from the subjective norm and 1 item from the descriptive norm). The reduction in items resulted in a new CFA, comprising 32 items grouped in 11 latent constructs, showing an acceptable model fit: $x^{2}$ = 1855.729, df = 403, *p* < 0.05, CFI = 0.945, TLI = 0.932, SRMR = 0.0479, RMSEA = 0.059, AIC = 2169.729. All factor loadings exceeded >0.4, except one FCPB item, which closely approached the cutoff, and it was therefore decided to keep it (0.390).

The initial TPB walking model comprised the same 11 latent constructs as the cycling model, and the initial model yielded the following results showing inadequate model fit:$\;x^{2}$ = 6624.397, df = 937, *p* < 0.05, CFI = 0.848, TLI = 0.832, SRMR = 0.0674, RMSEA = 0.077, AIC = 7004.397. Factor loadings ranged from 0.239 to 0.987. The same stepwise procedure following factor loadings and modification indices was conducted for improving the walking model [40], resulting in the reduction in the same items as the cycling model. The new CFA yielded the following results, showing an acceptable model fit: $x^{2}$ = 1969.532, df = 406, *p* < 0.05, CFI = 0.938, TLI = 0.924, SRMR = 0.0446, RMSEA = 0.061, AIC = 2277.532. Likewise, as in the cycling model, only one item did not meet the factor loading cutoff, but it did closely approach it (0.397).

The TPB constructs showed satisfying McDonald’s Omega and AVE values regarding both the cycling and walking model. All of the constructs correlating to another construct were lower than the root square AVE of each construct providing discriminant validity (not shown in the table) [44].

### 3.3. Parents’ Scores, Correlations and Means on the Various Components

Parent’s mean scores and item correlations on the various components are presented in “Table 3 and Table 4”. The fairly high mean values on positive attitudes and low mean values on negative attitudes indicate that parents were overall positive about letting their child cycle (M = 19.10 and 4.84) or walk (M = 19.08 and 5.96) to school. The rather high mean value for subjective and descriptive norms suggests that significant others would approve of it, and that significant others would let their child cycle (M = 5.93 and 5.10, respectively) and walk (M = 5.78 and 4.99, respectively) to school. The relatively high mean value on facilitating perceived behavioral control and the low mean value for impeding perceived behavioral control indicates that parents believed that it would be relatively easy to let their child cycle (M = 10.26 and 16.84) or walk (M = 10.03 and 16.80) to school. The fairly high mean value of intention demonstrated that a large proportion of parents intended to let their child cycle to school (M = 5.17). However, the intention to let the child walk was lower (M = 3.42). A rather high mean value on past cycling behavior (M = 4.37) indicated that a large proportion of parents had let their child cycle to school in the past three weeks. Relatively few had let them walk to school (M = 1.75). The low mean values for past behavior of car (M = 1.97) and bus (M = 1.35) indicted that few of the respondents reported that their child had used these travel modes in the past three weeks.

All TPB cycling and walking constructs and past cycling and walking behavior were significantly correlated with intention. Past cycling behavior, subjective norms and facilitating perceived behavioral control were most highly correlated with the intention to let the child cycle to school (r = 0.770, 0.595 and 0.591, respectively). Past walking behavior (r = 0.486), subjective norms (r = 0.364) and descriptive norms (r = 0.346) were most correlated with the intention to let the child walk to school. Impeding perceived behavioral control and negative attitudes were negatively correlated with the intention to let the child cycle (r = −0.288 and −0.492) and walk (r = −0.232 and −0.377) to school. Past behaviors of car and bus were negatively correlated with the intention to let the child cycle (r = −0.617 and −0.358) and walk (r = −0.361 and −0.206) to school.

However, contrary to the theory, the TPB constructs were not always more strongly correlated with intention than to each other. For parents’ intention to let the child cycle to school, subjective and descriptive norms were more correlated with each other than intention. Facilitating perceived behavioral control were more correlated with subjective and descriptive norms than intention. Additionally, subjective norms were more correlated with positive attitudes than intention.

Regarding parents’ intention to let the child walk to school, subjective and descriptive norms were more correlated with each other and positive attitude than intention. Facilitating perceived behavioral control were also more correlated with positive attitudes as well as subjective and descriptive norms than intention. Finally, descriptive norms and subjective norms answer options *“I do not know/Does not apply to me”* (treated as missing) for parents’ intention to let their child cycle to school were *n* = 301 and *n* = 133, respectively. Regarding walking to school, the same answer options were *n* = 307 and *n* = 144 (not shown in Table 3 and Table 4).

**Table 3 ijerph-18-11651-t003:** Means, standard deviations and zero-order Pearson correlations among the various cycling components.

Study Variable	M (SD)	1	2	3	4	5	6	7	8	9
1.INT ^†††^	5.17 (2.41)	-								
2.PATT ^†^	19.10 (5.68)	0.399 **	-							
3.NATT ^††^	4.84 (4.94)	−0.492 **	−0.147 **	-						
4.SN ^†††^	5.93 (1.66)	0.595 **	0.613 **	−0.394 **	-					
5.DN ^†††^	5.10 (1.69)	0.581 **	0.559 **	−0.324 **	0.771 **	-				
6.IPBC ^††^	10.26 (7.26)	−0.288 **	−0.046 ^n.s^	0.434 **	−0.241 **	−0.163 **	-			
7.FPBC ^††^	16.84 (5.89)	0.591 **	0.469 **	−0.381 **	0.670 **	0.661 **	−0.317 **	-		
8.PB Cycle ^††††^	4.37 (2.06)	0.770 **	0.245 **	−0.478 **	0.378 **	0.418 **	−0.317 **	0.445 **	-	
9.PB Car ^††††^	1.97 (1.72)	−0.617 **	−0.221 **	0.449 **	−0.380 **	−0.398 **	0.323 **	−0.397 **	−0.649 **	-
10.PB Bus ^††††^	1.35 (1.17)	−0.358 **	−0.133 **	0.393 **	−0.267 **	−0.204 **	0.323 **	−0.296 **	−0.383 ^n.s^	−0.001 ^n.s^

Note: ^†^ Scale 0.83–24.5 (higher values indicate a more positive attitude), ^††^ Scale 0.5–24.5 (higher values indicate a more negative attitude, more perceived impeding and facilitating beliefs), ^†††^ Scale 1–7 (higher values indicate stronger social norms and stronger intention to let the child cycle to school) ^††††^ Scale 1–6 (higher values indicate more frequently cycle, car, bus usage to school in the past). ** = *p* < 0.001; ^n.s^, non-significant. INT, intention; PATT, positive attitude; NATT, negative attitude; SN, subjective norm; DN, descriptive norm; IPCB, impeding perceived behavioral control; FPCB, facilitating perceived behavioral control; PB, past behavior.

**Table 4 ijerph-18-11651-t004:** Means, standard deviations and zero-order Pearson correlations among the various walking components.

Study Variable	M (SD)	1	2	3	4	5	6	7	8	9
1.INT ^†††^	3.42 (2.61)	-								
2.PATT ^†^	19.08 (5.70)	0.213 **	-							
3.NATT ^††^	5.96 (5.68)	−0.377 **	−0.135 **	-						
4.SN ^†††^	5.78 (1.79)	0.364 **	0.597 **	−0.416 **	-					
5.DN ^†††^	4.99 (1.72)	0.346 **	0.547 **	−0.309 **	0.730 **	-				
6.IPBC ^††^	10.03 (7.23)	−0.232 **	−0.050 ^n.s^	0.452 **	−0.287 **	−0.197 **	-			
7.FPBC ^††^	16.80 (5.87)	0.271 **	0.472 **	−0.310 **	0.629 **	0.622 **	−0.319 **	-		
8.PB Walk ^††††^	1.75 (1.47)	0.486 **	0.005 ^n.s^	−0.205 **	0.118 **	0.088 *	−0.141 **	0.035 ^n.s^	-	
9.PB Car ^††††^	1.97 (1.71)	−0.361 **	−0.219 **	0.446 **	−0.427 **	−0.409 **	0.322 **	−0.400 **	−0.146 **	-
10.PB Bus ^††††^	1.35 (1.17)	−0.206 **	−0.142 **	0.394 **	−0.320 **	−0.224 **	0.327 **	−0.301 **	−0.085 **	−0.001 ^n.s^

Note: ^†^ Scale 0.83–24.5 (higher values indicate a more positive attitude), ^††^ Scale 0.5–24.5 (higher values indicate a more negative attitude, more perceived impeding and facilitating beliefs), ^†††^ Scale 1–7 (higher values indicate stronger social norms and stronger intention to let the child cycle to school) ^††††^ Scale 1–6 (higher values indicate a more frequently walk, car and bus use to school in the past). ** = *p* < 0.001; * *p* < 0.05; ^n.s^, non-significant. INT, intention; PATT, positive attitude; NATT, negative attitude; SN, subjective norm; DN, descriptive norm; IPCB, impeding perceived behavioral control; FPCB, facilitating perceived behavioral control; PB, past behavior.

### 3.4. Parents Intention to Let Their Child Cycle or Walk to School

Estimations of parents’ intentions to let the child cycle or walk to school are presented in Table 5 and Table 6. A stepwise hierarchical linear regression analysis was used. The TPB constructs were entered in step one. Socio-economic background characteristics were entered in the second step. In the third step, past behavior was investigated, and finally, in the fourth step, the distance to school was entered. Separate analyses were conducted to test for multicollinearity and autocorrelation, showing that the level of tolerance was above 0. 2. The VIF was around 1–3 and the Durbin–Watson score was 1.980 (parents’ intention to the child cycle to school) and 1.869 (parents’ intention to let the child walk to school), indicating that this was not a problem [41].

The results from the regression analysis showed that the TPB explained 55.3% of the variance in parents’ intentions to let their child cycle to school (Table 5). In the first step, all of the original TPB variables showed up as important factors, except impeding perceived behavioral control, thus explaining variance in parents’ intentions. The most important factors were facilitating perceived behavioral control (β = 0.272, *p* < 0.001) and subjective norms (β = 0.147, *p* = 0.002). These were closely followed by positive attitudes (β = 0.140, *p* < 0.001) and descriptive norms (β = 0.128, *p* = 0.003). Negative attitudes had a highly negative impact on intentions (β = −0.286, *p* < 0.001). In step two, none of the socio-economic factors increased the variance, except for the academic school year, indicating that intentions increase as the child enters higher academic school years. When past travel behavior was entered in step three, the variance increased by another 18.3%, a highly significant increase. Past behavior of letting the child cycle to school had a great impact on intentions (β = 0.429, *p* < 0.001); meanwhile, past behavior of using a car had a negative impact (β = −0.177, *p* < 0.001). Interestingly, when past behavior was entered, descriptive norms and academic school year became insignificant, and impeding perceived behavioral control became significant. In the fourth step, the variance increased somewhat, showing that a shorter distance to school was associated with a higher intention. The full model explained 75% of the variance in parents’ intentions to let the child cycle to school.

**Table 5 ijerph-18-11651-t005:** Parents’ intention to let the child cycle to school. Hierarchical linear regression analysis.

	Step 1			Step 2			Step 3			Step 4		
	*B*	β	*p*	*B*	β	*p*	*B*	β	*p*	*B*	β	*p*
PATT	0.056	0.140	<0.001	0.057	0.142	<0.001	0.036	0.091	<0.001	0.036	0.089	<0.001
NATT	−0.134	−0.286	<0.001	−0.128	−0.273	<0.001	−0.029	−0.061	0.023	−0.024	−0.051	0.062
SN	0.199	0.147	0.002	0.226	0.166	<0.001	0.216	0.159	<0.001	0.211	0.155	<0.001
DN	0.180	0.128	0.003	0.141	0.100	0.021	0.039	0.028	0.401	0.044	0.031	0.344
IPCB	0.007	0.021	0.480	0.005	0.014	0.643	0.029	0.086	<0.001	0.034	0.100	<0.001
FPCB	0.107	0.272	<0.001	0.101	0.257	<0.001	0.070	0.179	<0.001	0.067	0.170	<0.001
Adj. R^2^		0.553	<0.001									
Gender of parent				−0.058	−0.010	0.715	−0.081	−0.014	0.501	−0.091	−0.015	0.446
Age of parent				−0.088	−0.045	0.108	−0.067	−0.034	0.109	−0.060	−0.031	0.143
Ethnicity of parent				−0.132	−0.012	0.650	−0.251	−0.022	0.259	−0.333	−0.029	0.135
Education of parent				0.278	0.051	0.058	0.194	0.036	0.082	0.184	0.034	0.097
Gender of child				0.133	0.028	0.281	0.035	0.007	0.708	0.049	0.010	0.598
School year				0.151	0.108	<0.001	0.007	0.005	0.829	0.022	0.015	0.496
Adj. R^2^					0.561	<0.001						
PB Cycle							0.503	0.429	<0.001	0.506	0.432	<0.001
PB Car							−0.243	−0.177	<0.001	−0.212	−0.155	<0.001
PB Bus							−0.098	−0.048	0.071	−0.055	−0.027	0.325
Adj. R^2^								0.747	<0.001			
Distance										0.118	0.078	0.003
Adj. R^2^											0.750	<0.001

Note: *B*, unstandardized coefficient; β, standardized regression coefficients; *p*, significance level; Adj. R^2^, explained variance in the dependent variable; PATT, positive attitude; NATT, negative attitude; SN, subjective norm; DN, descriptive norm; IPCB, impeding perceived behavioral control; FPCB, facilitating perceived behavioral control; PB, past behavior. Method = ENTER. Step 1, TPB constructs; Step 2, socioeconomic and demographic variables added; Step 3, PB added; Step 4, distance added.

The TPB explained 20.6% of the variance in parent´s intentions to let their child walk to school (Table 6). The most important factors were subjective norms (β = 0.125, *p* = 0.036) and descriptive norms (β = 0.122, *p* = 0.022). Negative attitudes had a highly negative impact on intentions (β = −0.206, *p* < 0.001). In the second step, the explained variance increased slightly, whereas parents’ gender and the child’s academic school year were significant. This indicates that the intention of letting the child walk to school is higher among mothers, and that the intention increases along with higher academic school year. In the third step, the explained variance increased by another 16.6% when past behavior was entered. A highly significant increase was induced by parent’s past behavior to let the child walk to school (β = 0.389, *p* < 0.001). Additionally, past behavior of using a car (β = −0.168, *p* < 0.001) had a significant negative impact on intention. Finally, when the distance to school was entered, the explained variance increased (β = 0.168, *p* < 0.001), indicating that a shorter distance to school is associated with greater intentions to let the child walk to school. Academic school year, parents’ gender, and past travel behavior (walk and car) remained significant in the full model. The full model explained 40.5% of the variance in parents’ intentions to let their child walk to school.

**Table 6 ijerph-18-11651-t006:** Parents’ intention to let the child walk to school. Hierarchical linear regression analysis.

	Step 1			Step 2			Step 3			Step 4		
	*B*	β	*p*	*B*	β	*p*	*B*	β	*p*	*B*	β	*p*
PATT	0.022	0.049	0.291	0.020	0.046	0.324	0.028	0.064	0.125	0.030	0.068	0.099
NATT	−0.096	−0.206	<0.001	−0.100	−0.215	<0.001	−0.043	−0.091	0.029	−0.028	−0.060	0.153
SN	0.174	0.125	0.036	0.218	0.157	0.008	0.139	0.100	0.058	0.086	0.062	0.241
DN	0.184	0.122	0.022	0.130	0.086	0.107	0.090	0.060	0.209	0.110	0.073	0.120
IPCB	−0.007	−0.020	0.632	−0.007	−0.019	0.637	0.012	0.032	0.395	0.021	0.055	0.142
FPCB	0.036	0.085	0.095	0.033	0.078	0.125	0.022	0.051	0.271	0.016	0.037	0.416
Adj. R^2^		0.206	<0.001									
Gender of parent				0.604	0.094	0.009	0.459	0.071	0.025	0.423	0.066	0.036
Age of parent				0.083	0.039	0.295	0.071	0.033	0.315	0.081	0.038	0.244
Ethnicity of parent				−0.713	−0.059	0.086	−0.164	−0.014	0.659	−0.365	−0.030	0.325
Education of parent				−0.212	−0.036	0.316	−0.130	−0.022	0.490	−0.160	−0.027	0.390
Gender of child				0.255	0.049	0.153	0.252	0.048	0.112	0.273	0.052	0.082
School year				0.148	0.097	0.009	0.124	0.081	0.021	0.155	0.102	0.004
Adj. R^2^					0.226	<0.001						
PB Walk							0.689	0.389	<0.001	0.661	0.373	<0.001
PB Car							−0.250	−0.168	<0.001	−0.198	−0.133	0.001
PB Bus							−0.190	−0.085	0.020	−0.107	−0.048	0.200
Adj. R^2^								0.392	<0.001			
Distance										0.274	0.168	<0.001
Adj. R^2^											0.405	<0.001

Note: *B*, unstandardized coefficient; β, standardized regression coefficients; *p*, significance level; Adj. R^2^, explained variance in the dependent variable; PATT, positive attitude; NATT, negative attitude; SN, subjective norm; DN, descriptive norm; IPCB, impeding perceived behavioral control; FPCB, facilitating perceived behavioral control; PB; past behavior. Method = ENTER. Step 1, TPB constructs; Step 2, socioeconomic and demographic variables added; Step 3, PB added; Step 4, distance added.

## 4. Discussion

The PILCAST questionnaire showed acceptable structural fit according to the TPB, and satisfying reliability, convergent and discriminant validity. Parents are the primary decision makers of children’s AST; therefore, it is of great importance to validate instruments which can identify the psychological antecedents of parents’ decisions to let their child cycle or walk to school when aiming to promote such behaviors. To the best of our knowledge, this is the first valid and reliable TPB-based questionnaire built on belief-based measurements, separated into cycling and walking in efforts to understand such intentions [21,22,23]. The questionnaire may therefore be important in efforts to develop a deeper and better understanding of these behaviors from a parental perspective, providing guidance when designing, implementing and evaluating interventions. In the regression analysis, the PILCAST questionnaire showed to be useful in explaining parents’ intentions because the TPB explained 55.3% of the variance in parents’ intentions to let their child cycle to school and 20.6% in parents’ intentions to let the child walk to school.

The results for parents’ intention to let their child walk to school are, however, somewhat lower compared to the previous average (39%) explained variance in intentions with reference to other behaviors [45]. However, our results are consistent with previous studies regarding parents’ intentions of letting the child walk to school, showing that the TPB explains about 20–27% of the variance in such intentions [22,23]. On the other hand, the amount of explained variance was substantially larger for parents’ intentions to let their child cycle to school, which was considerably above the average of explained variance compared to other behaviors [45].

### 4.1. Subjective and Descriptive Norms

Subjective and descriptive norms showed up as important factors concerning both parents’ intentions to let their child cycle and walk to school. These results would suggest that parents’ intentions increase if significant others would approve of parents letting their child cycle and walk to school, and if significant others would let their child cycle and walk to school. These results agree with previous research, where the most important factors determining parents’ intentions regarding their child walking to school were subjective [22,23] and descriptive norms [23]. However, these results are contrary to previous research regarding other PA-related behaviors. Subjective norms are often the construct providing the least explained variance compared to attitude and perceived behavioral control [45]. This indicates that AST behavior is somewhat distinguished from other PA-related behaviors [45]. A reason for subjective and descriptive norms showing up as important factors might be because these behaviors involve some kind of risk from a parental perspective [9]; thus, they are more attributable to social norms [22,23]. In their review, Rivis and Sheeran suggested that descriptive norms are more likely to be important when it comes to behaviors that involve risk [27]. In addition, our previous results from a qualitative study showed that being perceived as a “good parent” in the eyes of significant others was important to parents [33]. Parents explained that being a “good parent” could potentially be in conflict with AST, because it would include not exposing their children to risks, which consequently would lead to significant others considering them as irresponsible parents. On the other hand, the qualitative study also showed that if parents perceived that significant others allowed their child to use AST, they became more inclined to do the same [33]. Our results support those of Pang et al. [23], who concluded that it might be efficient to put effort into changing social norms to alter these kinds of intentions in parents.

### 4.2. Facilitating and Impeding Perceived Behavioral Control

The results support our suggestion that it would be beneficial not to focus on barriers only, but also facilitating factors, because the intention to let the child cycle to school increased if parents perceived it as relatively easy to do. Our analysis showed that parents perceive it rather easy to let their child cycle to school, because they felt that they can trust the child and the child’s ability to navigate in traffic, as well as when the environment is perceived as safe and there is access to separate walking and bicycle lanes. These results emphasize the importance of creating supportive environments for health-promoting PA behaviors such as AST [46]. Here, interventions might also play an important role, supporting both parental strategies (i.e., trusting the child and learning to navigate in traffic) [33] as well as developing children’s skills [47,48].

In the third step of the regression analysis, impeding perceived behavioral control became significant when past travel behavior was entered, concerning parents’ intentions to let the child cycle to school. These results suggest that parents’ intentions to let the child cycle to school increases with more perceived impeding factors, such as crossing unattended pedestrian crossings, crossing major roads, and traveling along roads with speeds higher than 40 km/h when the behavior is habituated. More studies are needed to validate these counterintuitive results. However, a possible reason for these results has been presented in previous studies showing that when AST behavior is habituated, parents are more inclined to negotiate barriers contrary to parents who do not let their child use AST [49,50]. Our results also showed that impeding and facilitating perceived behavioral control was correlated with parents’ intentions to let the child walk to school. However, the regressions analysis did not confirm these as important factors. This might be because walking to school does not require supportive environments and skills to the same extent as cycling [20]. This is, however, somewhat contrary to previous studies showing that perceived behavioral control is an important factor when it comes to parents´ intentions regarding their child walking to school [22,23]. On the other hand, Murtagh and colleagues found that perceived behavioral control did not predict behavior regarding children’s own perceptions about walking to school [30]. They argued that this predictor was not influenced by parents’ beliefs and might therefore need to be further investigated. Our results confirm that parents’ decisions regarding their child walking to school are not necessarily determined by facilitating or impeding perceived behavioral control.

### 4.3. Attitude

Holding a positive attitude explained parents’ intentions to let their child cycle to school. In this study, such attitudes referred to beliefs such as the child would improve their health, independency and concentration in school. The reason for a positive attitude not explaining parents’ intentions to let their child walk to school might be because parents were aware of the benefits, but the benefits were not sufficient to challenge the motive of their intention (decision) [28]. These results concur with previous studies showing that attitudes not provided explained variance in parents’ intentions regarding their child walking to school [22,23], even if the health benefits were well understood by parents [23]. This would also be somewhat in line with our previous results showing that parents do know that AST is beneficial, but occasionally need to prioritize faster transportation in order to make everyday living easier [33]. This also aligns with the results of the present study, showing that the past behavior of using the car to school had a negative impact on parents’ intentions to let the child walk and cycle to school as well as a negative attitude. The consequences of behaviors such as cycling and walking to school cannot be perceived as too cumbersome or taking too long time, otherwise parents might choose other transport modes. Consequently, walking to school might be a behavior that is more sensitive to the consequences of distance such as time taken than cycling, which is much faster and can cover further distances than walking [20]. This would also be consistent with our analysis showing that a shorter distance to school explained more variation in parents’ intention to let the child walk to school (β = 0.168, *p* < 0.001) than to let their child cycle to school (β = 0.078, *p* = 0.003). In the final step, when distance was entered into the regression, none of the TPB constructs remained independently significant concerning parents’ intention to let the child walk to school. The significance of distance to school has been reported previously to show that the TPB explained more variance in parents’ intentions regarding their child walking to school when the distances were shorter (<3 km) than when the distances were longer (>3 km) [23]. 

### 4.4. Past Behavior

When past behavior were entered into the regression analysis, the explained variance in parents’ intentions to let their child cycle to school increased substantially (18.3%), as well as the intention to let their child walk (16.6%). This result indicates that these behaviors might be under the influence of habituation. Ver Planken explains that when behavior is in “deliberate mode”, it is to a large extent, internally cued such as by people’s motivation, which follows the principle of the TPB [51]. However, when the behavior is in “habit mode”, it is mostly cued by external factors in the environment, such as people and places [51]. A shift from “deliberate mode” to “habit mode” would therefore have practical implications because the control of the behavior moves from the individual to the environment. Nevertheless, one previous study found that a change in the individuals’ context might open a window to disturbing automatic processes and activation of reasoned action, as a change in modal travel choice (less car use) was set in motion [52]. Similarly, Bamberg and colleagues argue that past behavior might be more important when the behavior is stable [31]. They found that past behavior lost its effect in the post-intervention analysis, when the motives of students’ school transport were challenged with free bus tickets. This implies that what could be considered as a relatively habituated behavior such as transport to school, in its sense, is not purely automatic [31]. Even when the behavior is routine, complex social behavior (i.e., modal travel choice) seem to be regulated at some level even if the level of conscious awareness is low (i.e., derived by internal cues) [31]. As a consequence, even minor events could disrupt automatic processes and aggregate reasoned action [31]. Thus, in line with Murtagh and colleagues’ suggestion it might be beneficial to incorporate AST interventions when natural process of change occur in the school context [30]. In Sweden, several contextual changes occur during the school years (i.e., from pre-school to primary school and high school to secondary school), making these transitions suitable for the incorporation of interventions. Our results indicate that when aiming to promote AST with the aid of interventions, one should focus on both internally and externally cued determinants to alter these intentions in parents.

### 4.5. Strengths and Limitations

The major strength of the present study was the use of belief-based measures providing some interesting results and insights, contributing to previous knowledge about parents’ intentions regarding AST. There are, however, some limitations worth mentioning. First, both parents could respond to the web questionnaire and submit one unique questionnaire based on each child. This might have influenced the representativeness of the present sample in the municipality. However, our decision to let both parents respond based on each child was based on previous studies highlighting the limitations of studies only collecting data from one parent per household [10] and because mothers and fathers tend to individually assess of their children concerning AST [33,53]. Secondly, the gender distribution in this sample was skewed. This may have influenced the results, which therefore should be interpreted bearing this in mind. Nevertheless, the gender distribution was quite similar to recent studies within this research field [10,22,23]. However, as suggested by others, future research would benefit from data with a more equal gender distribution [10]. Thirdly, regarding descriptive and subjective norms, a relatively large proportion of parents used the answer the option *“I do not know/Does not apply to me*”. This result would imply that these questions were harder to answer than expected. This has, however, been seen elsewhere [54]. Nevertheless, future studies may overlook these questions and answer options. Additionally, although the PILCAST questionnaire showed satisfactory validity and reliability in this sample, external validity should be further addressed in other settings and samples [55]. Finally, actual behavior was not measured in this study; therefore, future research could include objective measures of these behaviors. A final limitation of this study is related to the ongoing project to promote AST in the region [39], which—along with a large number having a short distance to school—might explain the high number of children who cycled or walked to school in this sample. In efforts to not make the present study too extensive, this effect will be further investigated in future studies.

## 5. Conclusions

The PILCAST questionnaire has been shown to be a valid and reliable instrument to understand parents´ intentions to let their child cycle or walk to school. The regression analyses showed that the TPB explained 55.3% of the variance in parents’ intentions to let their child cycle to school, and 20.6% of parents’ intentions to let their child walk to school. The most influential TPB constructs regarding parents’ intention to let the child cycle to school were facilitating perceived behavioral control, positive attitudes, subjective norms and descriptive norms, and regarding walking it was subjective and descriptive norms. When adding past behavior, the explained variance increased by a further 18.3% (cycling) and 16.6% (walking). Together, these results suggest that parents’ intentions to let their child cycle or walk to school is influenced by both belief-based and habituated processes, and that it might be beneficial to target these behaviors (i.e., parents’ intentions to let the child cycle or walk to school) differently. The PILCAST questionnaire contributes to a better understanding of the psychological antecedents involving parents’ decisions to let their child cycle or walk to school, and may therefore provide guidance when designing, implementing and evaluating interventions aiming to promote AST.

## Data Availability

The data presented in this study are not openly available because participants did not provide informed consent for data sharing.

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
