# Peer review of "Development and Initial Validation of the PILCAST Questionnaire: Understanding Parents’ Intentions to Let Their Child Cycle or Walk to School"

_ijerph, 2021, doi:10.3390/ijerph182111651_

Round 1

Reviewer 1 Report

I would like to thank the authors and editor for giving me the opportunity to review this interesting as well as important paper. I believe that more interventions are needed to promote PA in children and active school transportation is one of them.

My general comments are related to one of the limitations and is the difference in gender of the population that responded to the questionnarie. I think that when explaining the results (example page 10, line 353) that having 79% vs 20 % gender differences in the participants, maybe the authors should be careful in the explaing the results.

There is also a typo in page 3 line 96. The sentence does not read well.

Author Response

Revised Manuscript

Dear Editor,

We are thankful for the opportunity to improve the manuscript “Development and initial validation of the PILCAST questionnaire: Understanding parents´ intentions to let their child cycle or walk to school” for publication in International Journal of Environmental Research and Public Health, special issue Promotion of Active commuting to school. We appreciate the helpful comments received from the reviewers. Below we have explained (in italic text) what we have re-written in light of these comments. In the revised manuscript the changes are highlighted.

  1. My general comments are related to one of the limitations and is the difference in gender of the population that responded to the questionnarie. I think that when explaining the results (example page 10, line 353) that having 79% vs 20 % gender differences in the participants, maybe the authors should be careful in the explaing the results.

We agree and have added text about this in the discussion section.

  1. There is also a typo in page 3 line 96. The sentence does not read well.

We have deleted the typo in this sentence.

We are grateful to be able to strengthen our paper guided and inspired by your helpful comments and suggestions, and we hope the improvements will prove to be satisfactory.

On behalf, of the first author Hanna Forsberg and coauthors Anna-Karin Lindqvist, Sonja Forward, Lars Nyberg and Stina Rutberg.

Reviewer 2 Report

This study aimed to develop and validate a questionnaire that measures the constructs of the parents’ intentions to let their child cycle or walk to school, based on the Theory of Planned Behavior (TPB). The authors performed a four-stage study, including a suitable methodology to develop the questionnaire and a sampling of 1024 responses to validate the items/constructs. The results showed that the scales based on the TPB explained 55.3% and 20.6% of parents’ intentions to let the child cycle or walk to school, respectively. The study is relevant because the measured constructs are pillars of physical activity strategies for pupils. However, methodological aspects were not suitable in the study or were not reported in the text; thus, they imply issues on the internal and external validity of the study and the reporting quality. Please see the specific comments.

 Abstract:

  1. Line 9: Please check the sentence to clarify the message: “Children’s levels of physical activity (PA) do not meet the recommendations”.
  2. Please add information on the structure of the questionnaire: how many items? How many subscales or specific constructs was the instrument organized?
  3. Lines 21-22: the conclusion of the abstract should be based on the aim of the study (validation of the questionnaire) instead of intervention elements. Please adjust it.

Introduction:

  1. Line 27: This is the first time that “physical activity” is shown in the text. Please add the abbreviature appropriately.
  2. First paragraph: the readability of the introduction would be improved with a change in the structure of the paragraph. For instance, I suggest that the content be organized into two different paragraphs.
  3. The introduction has a good rationale for the theoretical bases of the proposed questionnaire. However, information on the relevance of the new questionnaire and limitations on the previous questionnaires were not considered. Why did the authors develop a new questionnaire? Are there other questionnaires or not? Are they limited to measure the construct? This was addressed in the discussion only. Please add information on the relevance of the particular study in the introduction.
  4. Line 82: “To date, few studies have used the TPB to assess parent’s intention to let their child cycle or walk to school.” Please add a reference to support the sentence.

Method:

  1. Lines 92-95: the first stage of the study aimed to develop the questionnaire, based on a qualitative approach. However, it is unclear how information was used to develop the first version of the questionnaire. How many items/constructs were considered for the second stage?
  2. Lines 96-103: During the transformation of qualitative content into items of the questionnaire, it is important to consider the previous expertise of the research team in the development of questionnaires. Has the research team who performed the second stage previous experience in this process? Please describe it.
  3. Was there any reference to support the building of the items (reference on development of questionnaires per si)?
  4. Was there any sample size estimation a priori? How did the authors decide to stop the data collection? Please clarify it.
  5. Lines 194-196: Please clarify how the scales were considered in the correlation and regression analyses. Were they estimated by CFA or based on crude (sum) of the scores from the items?
  6. Please add information on the descriptive variables that were shown in Table 1, but they were not described in the methods.

Results:

  1. I suggest adding supplementary material with each item included in the questionnaire according to scale and with information on factor loading of each item. This is important to detail the questionnaire for researchers that may be interested in using the instrument further.
  2. Table 2: Were the values of the Table obtained before or after (final values) the steps to fit the model (excluding items, etc.)? Please clarify it. Also, if data from Table 2 represent final values, I suggest adding supplementary material with initial data.
  3. Please add information in the methods on other variables besides TPB, i.e., parent´s mean scores, walking behavior, and others.
  4. Tables 5 and 6: Please describe the steps of the hierarchical linear regression analysis in the footnotes of the Tables.

Discussion

  1. Lines 377-378: I understand the authors’ assertion on the questionnaire may be helpful to intervention studies, however, questions from observational may also be answered by using this questionnaire. How observational studies can use the questionnaire? Please clarify it.
  2. I suggest adding information on the research and practice implications with the new study. Could the questionnaire be used in research on this topic? How? Could practitioners use the questionnaire for active transportation promotion strategies? How?

Conclusion:

  1. The conclusion should be focused on the study’s aim, i.e., the development and validation of the questionnaire. The assertions were focused on the associations in general.

Author Response

Revised Manuscript

Dear Editor,

We are thankful for the opportunity to improve the manuscript “Development and initial validation of the PILCAST questionnaire: Understanding parents´ intentions to let their child cycle or walk to school” for publication in International Journal of Environmental Research and Public Health, special issue Promotion of Active commuting to school. We appreciate the helpful comments received from the reviewers. Below we have explained (in italic text) what we have re-written in light of these comments. In the revised manuscript the changes are highlighted.

Reviewer 2

Abstract:

  1. Line 9: Please check the sentence to clarify the message: “Children’s levels of physical activity (PA) do not meet the recommendations”.

We have added and changed the text in the abstract section.

  1. Please add information on the structure of the questionnaire: how many items? How many subscales or specific constructs was the instrument organized?

Yes, we agree that this needs to be clarified and we have added text about this in the abstract section.

  1. Lines 21-22: the conclusion of the abstract should be based on the aim of the study (validation of the questionnaire) instead of intervention elements. Please adjust it.

Thank you for the valuable comment, we have rewritten, adjusted and added text in the abstract section. 

Introduction:

  1. Line 27: This is the first time that “physical activity” is shown in the text. Please add the abbreviature appropriately.

Thank you for making us aware about the misplacing of this abbreviation. We have changed this.

  1. First paragraph: the readability of the introduction would be improved with a change in the structure of the paragraph. For instance, I suggest that the content be organized into two different paragraphs.

We agree and we have added a second paragraph in the introduction section.

  1. The introduction has a good rationale for the theoretical bases of the proposed questionnaire. However, information on the relevance of the new questionnaire and limitations on the previous questionnaires were not considered. Why did the authors develop a new questionnaire? Are there other questionnaires or not? Are they limited to measure the construct? This was addressed in the discussion only. Please add information on the relevance of the particular study in the introduction.

Thank you for this important comment and yes we agree. To clarify this we have moved text from the discussion as well as added text in section “Theoretical framework”.

  1. Line 82: “To date, few studies have used the TPB to assess parent’s intention to let their child cycle or walk to school.” Please add a reference to support the sentence.

We have added references, rewritten, and moved this sentence to the section “Theoretical framework”.

Method:

  1. Lines 92-95: the first stage of the study aimed to develop the questionnaire, based on a qualitative approach. However, it is unclear how information was used to develop the first version of the questionnaire.

Thank you for the comment. We have clarified the use of the qualitative study and that it was in line with the guidelines of Ajzen Icek. 

How many items/constructs were considered for the second stage?

Thank you for pointing this out. We have added text to clarify this in the method section, “Procedure and measures”.

  1. Lines 96-103: During the transformation of qualitative content into items of the questionnaire, it is important to consider the previous expertise of the research team in the development of questionnaires. Has the research team who performed the second stage previous experience in this process? Please describe it.

We appreciate your comment, and we have made a clarification about this by adding text in the method section, “Procedure and measures”.

  1. Was there any reference to support the building of the items (reference on development of questionnaires per si)?

We followed the guidelines from Ajzen Icek on how to develop an extended version of a TPB questionnaire, which we refer to in the method section, “Procedure and measures” reference 32 and also 27 which give further support in adding the descriptive norm construct.

  1. Was there any sample size estimation a priori? How did the authors decide to stop the data collection? Please clarify it.

We have clarified this remark by adding text in the method section, “Procedure and measures”.

  1. Lines 194-196: Please clarify how the scales were considered in the correlation and regression analyses. Were they estimated by CFA or based on crude (sum) of the scores from the items?

In the light of your helpful comment we have acknowledged the need of a clarification and corrected a misplacing of text in the result section describing steps of analysis. Therefore, we have added and rewritten text in the data analysis section, as well as moved some text from the result section and hopefully this will clarify this issue.   

  1. Please add information on the descriptive variables that were shown in Table 1, but they were not described in the methods.

Thank you for pointing this out, we have added text about the descriptive variables in Table 1 in the method section.

Results:

  1. I suggest adding supplementary material with each item included in the questionnaire according to scale and with information on factor loading of each item. This is important to detail the questionnaire for researchers that may be interested in using the instrument further.

We agree with your comment that the factor loadings of each items is important and have added all of the factor loadings for each item in the constructs, in the results section Table 2, and removed “factor loading range”.

  1. Table 2: Were the values of the Table obtained before or after (final values) the steps to fit the model (excluding items, etc.)? Please clarify it. Also, if data from Table 2 represent final values, I suggest adding supplementary material with initial data.

Thank you for this comment, we have added text in the results section, clarifying that the values in Table 2 are the final values. Initial data is presented for the first TPB cycling and walking model covering model fit indices and the range of factor loadings under the sub heading “Construct validity and reliability”. We would prefer to keep it this way.

  1. Please add information in the methods on other variables besides TPB, i.e., parent´s mean scores, walking behavior, and others.

We are not sure if we have understood this remark correctly. In the added text, in the data analysis section we hope this was resolved.  

  1. Tables 5 and 6:Please describe the steps of the hierarchical linear regression analysis in the footnotes of the Tables.

We agree to your comment and we have also added text that describes the steps of the regression analysis in the footnotes of Table 5 and 6

Discussion

18/19. Lines 377-378: I understand the authors’ assertion on the questionnaire may be helpful to intervention studies, however, questions from observational may also be answered by using this questionnaire. How observational studies can use the questionnaire? Please clarify it. I suggest adding information on the research and practice implications with the new study. Could the questionnaire be used in research on this topic? How? Could practitioners use the questionnaire for active transportation promotion strategies? How?

We appreciate your insightful remarks and we have rewritten and added text in the beginning of the discussion.

Conclusion:

The conclusion should be focused on the study’s aim, i.e., the development and validation of the questionnaire. The assertions were focused on the associations in general.

Thank you for acknowledging this important issue. We have looked over, removed text about associations and rewritten the conclusion.

We are grateful to be able to strengthen our paper guided and inspired by your helpful comments and suggestions, and we hope the improvements will prove to be satisfactory.

On behalf, of the first author Hanna Forsberg and coauthors Anna-Karin Lindqvist, Sonja Forward, Lars Nyberg and Stina Rutberg.

Round 2

Reviewer 2 Report

Thank the editors for the invitation to evaluate the manuscript and the authors for the revised version. I believe the authors addressed the suggestions and I believe the manuscript could be approved for publication.